# Evolutionary Mechanisms of the Emergence of the Variants of Concern of SARS-CoV-2

**DOI:** 10.3390/v17020197

**Published:** 2025-01-30

**Authors:** Igor M. Rouzine

**Affiliations:** Sechenov Institute of Evolutionary Physiology and Biochemistry, Russian Academy of Sciences, St. Petersburg 194223, Russia; igor.rouzine@iephb.ru

**Keywords:** immunosuppressed patient, chronic infection, primary mutation, compensatory mutation, recombination

## Abstract

The evolutionary origin of the variants of concern (VOCs) of SARS-CoV-2, characterized by a large number of new substitutions and strong changes in virulence and transmission rate, is intensely debated. The leading explanation in the literature is a chronic infection in immunocompromised individuals, where the virus evolves before returning into the main population. The present article reviews less-investigated hypotheses of VOC emergence with transmission between acutely infected hosts, with a focus on the mathematical models of stochastic evolution that have proved to be useful for other viruses, such as HIV and influenza virus. The central message is that understanding the acting factors of VOC evolution requires the framework of stochastic multi-locus evolution models, and that alternative hypotheses can be effectively verified by fitting results of computer simulation to empirical data.

## 1. Introduction

Similarly to HIV, HCV, and influenza virus, SARS-CoV-2 is perpetually acquiring new substitutions in its genome, with a very high mean substitution rate of (0.6–1.6) · $10^{-3}$/year/site, depending on the subtype [1,2] (Figure 1). The substitution rate of its envelope protein, Spike, is even faster, at (5–6) · $10^{-3}$/year/site, and second only to the envelope protein of HIV [3]. Millions of genomes have been sequenced to date, producing a vast amount of data on genetic variation, antigenicity, and transmission [4,5,6,7,8]. The variants of concern (VOCs) of SARS-CoV-2, such as Alpha (B.1.1.7), Delta (B.1.617.2), and Omicron (B.1.1.529 and its subvariants B.A1, B.A2, etc.), particularly stand out, due to their changes in virulence and transmission rate. For example, the variant Alpha had at least 43% faster transmission than the previous virus variant [9]. A VOC carries a new set of a dozen or more mostly non-synonymous mutations [10,11,12,13]. Phylogenetic analysis demonstrates that a VOC does not descend directly from any of the current dominant virus variants, but represents a sister strain, with a common ancestor that existed many months prior [10,11,12,13]. The evolutionary cause of this phenomenon remains unknown. Understanding VOC genesis demands a concerted effort of clinicians, epidemiologists, virologists, immunologists, and virus evolution experts.

## 2. Epidemiological Reservoirs (Silent Spread)

One possible explanation postulates the existence of a hidden epidemiological reservoir, where the virus evolves un-noticed and then returns to the population. Such a reservoir could be an animal population (reverse zoonosis) [21,22] or a human population not covered by genetic surveillance [6,13,23,24]. The two hypotheses are similar. The facts in favor of them include numerous documented cases of a pathogen passing between an animal reservoir and a human population (e.g., bubonic plague, HIV, influenza), as well as certain SARS-CoV-2 variants being found only in specific locations, at least early on following their first detection [24]. A difference between the animal reservoir hypothesis and the hypothesis of silent spread in a human population is the presumed evolutionary rate. Another difference is that cross-species jumps often require additional adaptation mutations. The silent spread hypothesis could be confirmed by tracing a VOC to a specific human or animal reservoir.

## 3. Evolution in Immunocompromised Individuals

Much discussed in the literature is the hypothesis that VOCs may evolve within immunocompromised individuals who exhibit a long-term chronic infection [10,25,26,27,28,29,30]. Due to its adaptation to a human host in the presence of a weakened immune response, the virus could rapidly accumulate many mutations and then return to the general population. The prevalence of chronic infections in the population is small, 0.1–0.5% [31], but they could seed new highly fit variants. In the support of this idea, evolution in immunocompromised patients and VOCs share common features, including some hallmark mutations in the Spike protein and convergent evolution [10]. Several studies have found an accelerated substitution rate within some chronic individuals compared to the substitution rate of the virus in the main population [25,26,27,28,29,30], although some have challenged this [2]. However, acceleration is not specific evidence for seeding VOCs from these patients. The speed of viral evolution is often faster in chronically infected individuals due to a faster turnover; this is a common phenomenon among viruses. These data are also consistent with viral adaptation occurring in parallel and independently at two biological levels. One level is the evolution within a chronic patient, and another level is the evolution due to transmission in the general population (in hosts with acute infections). Both processes confer viral adaptation to human species, and likely select for many (but not all) similar substitutions.

A search for drivers of VOC evolution in 27 chronic patients demonstrated that, although substitutions in chronic patients include some hallmark VOC mutations, the long subsets of mutations associated with the global transmission of VOCs are absent from these patients [32]. The difference between VOCs and the virus in chronic patients could be explained by an evolutionary conflict between two biological levels [32,33]: the level of a chronically infected host and the level of a population. The evolutionary conflict, however, does not require chronic infection, and can also result from the parallel evolution at the two biological levels. Nevertheless, the chronic patient hypothesis remains the leading hypothesis in the literature, and deserves further investigation. Identification of the source patients for at least some VOCs would allow direct confirmation. Recently, shedding of novel SARS-CoV-2 Spike sequences to wastewater was traced to a single commercial building, and has been detected and interpreted as coming from a chronically infected individual [34].

## 4. Evolutionary Forces in the General Population

The above class of potential explanations was based on virus evolution in and seeding from a special reservoir of population, other than the general population. Another direction of the search is the detailed analysis of biological mechanisms shaping virus evolution in the general population. In principle, the same forces can also act in reservoirs, but the presence of reservoirs is not important for this line of inquiry. Viruses evolve due to a complex interplay between various biological factors, including random mutation, natural selection, epistatic interaction, random genetic drift, linkage between loci due to common phylogeny, recombination, migration, and inhomogeneous population structure (Box 1). The interplay between these evolutionary factors, which can be readily simulated in silico, generates a set of specific evolutionary patterns, which can be compared to genomic data in the public domain. All hypotheses of this class are testable, and their confirmation or falsification would help with understanding the origins of VOCs. Two examples follow.

Box 1Terminology.Allele—a variant of a locus.Compensatory mutation—a mutation at another locus, compensating the deleterious effect of a mutation at the given locus.Convergent evolution—evolution towards the same genetic sequence in independent populations.Fitness (reproduction number)—the average number of cells infected by a virus from one infected cell, or the average number of individuals infected by one individual. Fitness valley—a sequence of mutations with a transient fitness decrease in the middle.Genetic drift—random fluctuation in the progeny number of a virus genome, measured by the number of infected cells for within-host evolution and the number of infected hosts for population-level evolution, respectively. It causes random fluctuations in allelic frequency over time, and may cause their eventual loss.Genetic linkage—co-inheritance of alleles at different loci because they are close together on a chromosome (i.e., low chance of recombination splitting them up)Immune escape—a mutation causing a loss of recognition.Locus—a small variable part of a genome (usually, a nucleotide or amino acid position)Natural selection—an increase in the number of genetic variants with a larger fitness over generations.Recombination—a mosaic mix of two individual genomes, due to them switching templates during replication.

## 5. Fitness Valley Hypothesis

VOCs may arise early on, while still undetectable, due to a cascade of compensatory mutations emerging in a population after a primary mutation, conferring a major change in viral properties [35]. A population-level simulation study proposed this effect in conjunction with the evolution in immunocompromised patients [35]. However, under certain conditions, a fitness valley could also be crossed in the general population, in direct patient-to-patient transmission. The authors of [36] used computer simulation to interpret the origin of VOCs using simple fitness landscapes (a single mutation, several additive mutations, and a fitness plateau followed by a single mutation). The fitness valley, in contrast, includes a transient decrease in fitness forced by some changes in the environment (tissue, host, species, immune response). This “fitness valley” effect has been previously observed or inferred for various viruses, including HIV [37,38] and influenza [39,40], and studied using mathematical models [40,41]. For example, a primary mutation can help a virus to escape the immune response, adapt to a species or a tissue, or expand the pH range in which it can replicate. Mutations conferring these benefits to a virus, but reducing its overall fitness, are followed by compensatory mutations that elevate its fitness above the initial level.

Compensatory mutations (positive epistasis) have also been observed for SARS-CoV-2. Mutations in neutralizing antibody-binding regions that help SARS-CoV-2 to escape immune recognition during transmission [3,4,6,20,42] negatively affect the function of receptor binding, and may also have to be compensated for elsewhere. K417N and E484K in Spike seem to help it to avoid antibody recognition, but they also remove two salt-bridges with ACE2 (the target receptor on our cells); however, N501Y appears to increase affinity for ACE2, and might compensate for the loss of the salt-bridges [43]. The tradeoff between antibody escape and transmissibility has been inferred in immunocompromised patients [32]. The same is likely true for mutations conferring escape from innate responses to SARS-CoV-2 [44,45].

A classic example of the fitness valley effect is the rapid formation of drug resistance in HIV in patients treated with early antiretroviral inhibitors, where the initial primary mutation decreasing the viral sensitivity to a drug is followed by many compensatory mutations [46,47]. For example, HIV-1 develops resistance to protease inhibitors due to a protease mutation that reduces binding affinity [38,48,49]. The fitness cost of drug resistance is high, and requires one or more additional mutations in protease to compensate for reduced fitness in order for resistance to evolve. Compensatory cleavage-site mutations in the capsid protein are required as well [50].

Another example is mutations in the HIV genome that help it to escape the cytotoxic immune response in a host [51,52] (Figure 2A). The viral replication capacity lowered by an escape mutation is later gradually restored by compensatory mutations [53,54,55]. Compensatory alleles with the highest degree of compensation grow the fastest, creating a cascade of compensatory mutations over time. This process is thought to be responsible for the high diversity and rapid HIV evolution in untreated patients [56]. The loss of fitness due to escape mutations, and the partial loss of recognition by the immune system, together determine the order, the number, and the evolution rates of escape mutations [57,58]. Compensatory mutations prevent the reversion of escape mutations after the decay of CD8 T cell clones against escaped epitopes. Bacteria developing antibiotic resistance also pass through such fitness valleys [59].

To test the “fitness valley” hypothesis, one has to measure the network of epistatic interactions responsible for the transition between the initial and final viral strains. The fitness landscape comprising the epistatic network, and the costs of separate mutations, can be inferred from genetic samples using various methods [40,60,61,62,63,64]. The methods that take into account the effects of genetic linkage are more reliable. The signature of a fitness valley is a simple epistatic network, with a shape close to that of a star, and approximately one path connecting any two nodes (connectivity close to 1). Indeed, most epistatic connections exist due to compensation for the same primary mutation. Tertiary mutations compensating for the compensatory mutations can add some loops to the star, raising the connectivity slightly above 1. In one example, to explain the origin of the 2009 pandemic of Influenza A H1N1, the epistatic network was inferred for the neuraminidase protein [40] (Figure 2B). The epistatic network has a mostly a star-like appearance due to direct compensatory mutations, with a few loops due to secondary compensations. The primary mutation 248 is well known to decrease the sensitivity of the virus to pH [65]. Observation of a similar network for SARS-CoV-2 would represent evidence in favor of the “fitness valley” hypothesis. At the present moment, the observation of strong compensation in SARS-CoV-2 [43] represents indirect evidence, because it proves that compensatory mutations required for the fitness valley effect, in principle, exist for this virus.

Another test of the “fitness valley” hypothesis is that locations of the inferred compensatory residues must follow a structural pattern that is consistent with a specific mechanism of compensation. In this example, all compensatory sites are located on the protein surface in alpha helixes. Apparently, they serve to re-adjust the angles between beta-sheets that are changed by the primary mutation 248, representing a case of allosteric interaction (Figure 2C).

## 6. Multi-Locus Evolution with Recombination

Variants with large sets of newly acquired mutations could also arise due to multiple recombination events, with fitter recombinants then rising in frequency through natural selection. Indeed, SARS-CoV-2 has observable crossover recombination due to co-infection events [66,67,68,69]. In this hypothesis, highly fit sequences are formed, when still at undetectable levels, by rare recombination events (Figure 3A). This possibility is supported by computational and mathematical models, that have not yet been applied to SARS-CoV-2, which take advantage of the fact that, due to random recombination crossovers, some rare progeny genomes can have an unusually large number of beneficial alleles compared to their parents [70,71,72,73]. These models demonstrate that the evolution of a viral population can be understood as a distribution of genomes in fitness traveling along the fitness coordinate. The front of the distribution moves forward towards a higher fitness, because these rare recombinants, with an excess of beneficial alleles, are the most fit in a population [70]. Starting from a single copy, although not detectable initially, such recombinant lineages eventually outgrow all others, and become a new dominant strain in a population (Figure 3A). The time delay with which this process occurs, and the rapid growth of the lineage, could be an explanation for the “sudden” emergence of VOCs. Interestingly, a computer simulation based on such a model has generated a polyphyletic tree and a distribution of beneficial mutations among branches (Figure 3B) resembling that of a VOC [74] (Figure 3C). Initially, the genetic distance from the original virus strain increased linearly with time. When the variant Alpha appeared in the autumn of 2021, it deviated from this gradual increase by exhibiting a jump by 10 distinct mutations, including three short deletions, one knockout, and six amino acid replacements (Figure 3C). Phylogeny hinted at a common ancestor of Alpha (B.1.1.7) and the previous strain (20E or B.1.177) occurring sometime around March. The simulation (Figure 3B) reproduced these features qualitatively, including independent subtrees at different and even the same moments of time, and jumps in the mutation number. For the simulation details, see the caption of Figure 3 and [74].

The simplest way to rule out the recombination hypothesis is to demonstrate, with the use of a computer simulation, that the recombination incidence of SARS-CoV-2 is too low to cause such a frequent emergence of VOCs. The unknown input parameter is the effective outcrossing rate, defined as the probability that a genome undergoes recombination with another genome. This quantity depends on the effective rate of co-infection with two viral variants [71,73]. The last value is unknown, and can be either higher or lower than the average rate of individual co-infection, because transmission rate varies strongly among individuals for a variety of reasons, including superspreading events [76,77,78].

Fortunately, there are methods of inference of the outcrossing rate from genomic data. For example, the outcrossing rate in an average untreated HIV-infected individual was demonstrated, by two independent methods, to be around 1.5% [79,80]. Such techniques could be re-applied to SARS-CoV-2 data. If the estimated outcrossing rate is too low, and a Monte Carlo simulation based on it does not predict the emergence of VOCs with a frequency similar to that observed in real data, the hypothesis is falsified. This would not mean, of course, that recombination events do not occur, but only that they are insufficient to explain the emergence of the VOC under investigation. If, however, it is quantitatively consistent with the observed emergence, the hypothesis will be elevated to the rank of a working model. The discovery of the two parental strains of a VOC would provide its ultimate confirmation, and bioinformatics tools could help with wide screening.

Note that multi-locus models of evolution predict that the fittest strains are formed by sequential multiple recombination events [70,73,74], which may be difficult to detect. Genetic evidence demonstrates the importance of recombination in the formation of recent genetic variants of SARS-CoV-2 [81,82,83,84,85]. However, the fact that a new strain is a recombinant does not prove that recombination is the primary driver of their emergence, but only that it participates in its formation. The most important evolutionary factor can be hiding behind the scene. Only by performing computer simulation for alternative dynamic models, and comparing their predictions to genomic data, can it be determined which evolutionary mechanism is the primary driver [56,79,86,87,88].

The above hypotheses are not mutually excluding. For example, fitness valley crossing or recombination may occur within chronically infected patients, and then transfer to the population. A simulation study at the level of a population [35] has found that, when the pathogen does not have to cross a fitness valley for immune escape to occur (no epistasis), this has no principal effect on antigenic evolution in immunocompromised individuals. However, if a fitness valley exists at the between-host level (epistasis), persistent infections of immunocompromised individuals allow mutations to accumulate in a population. Usually, a fitness valley is crossed under a change of conditions, such as in the presence of an immune response or an antiviral, or during transmission between species, individual hosts, or organs; these factors need to be investigated.

An interesting question is whether the evolutionary pressure from infection prevention and control (lockdowns, vaccines, masks) affects the hypotheses presented in this manuscript. Because these hypotheses are poorly studied in the context of SARS-CoV-2, and we do not know which one is true, only future simulation and data analysis will be able to answer this question.

## 7. Conclusions

To summarize, silent evolution in population reservoirs or chronically infected patients is not the only possibility that one should focus on in the search of explanations for VOCs. The explanations based on the interplay between different factors driving virus evolution in the general population, such as a fitness valley and multi-locus evolution with recombination, which have received much less attention in the literature on SARS-CoV-2 but are well studied for other viruses, deserve more careful investigation. Although these explanations do not require evolution in chronically infected patients, they are compatible with it. Once one of these hypotheses about evolutionary mechanisms is confirmed, as described above, and the relevant parameters are estimated, the probabilistic prediction of future VOCs in the next few years, by Monte Carlo simulation, will become possible.

## Figures and Tables

**Figure 1 viruses-17-00197-f001:**
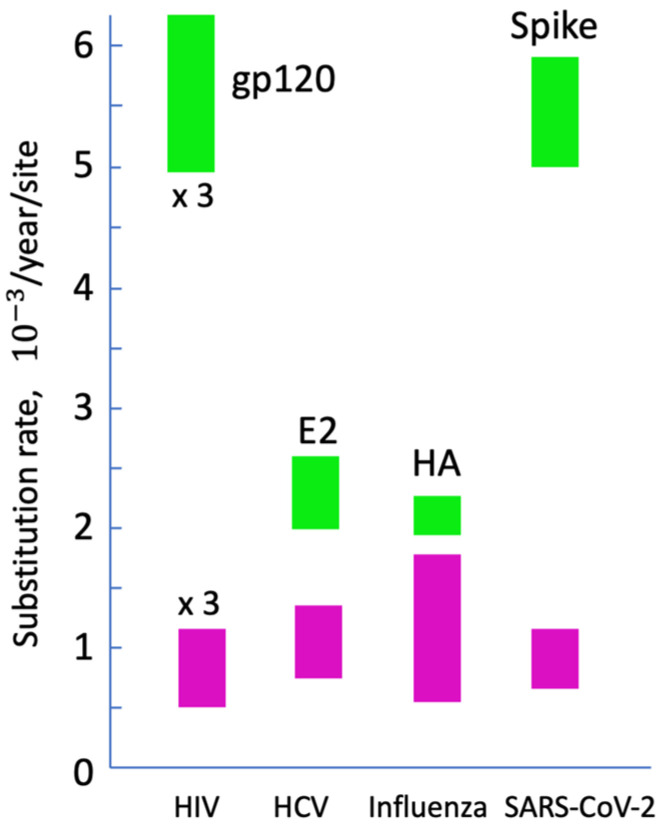
Viral substitution rates. The purple rectangles show the intervals of the median values for the most rapidly and most slowly evolving subtypes of HIV, HCV, influenza virus, and SARS-CoV-2 for the full genome [1,14,15,16]. The green rectangles correspond to the envelope proteins targeted by neutralizing antibodies [3,17,18,19]. To fit into the plot, HIV values are shown at a third of their actual value. Based on [20].

**Figure 2 viruses-17-00197-f002:**
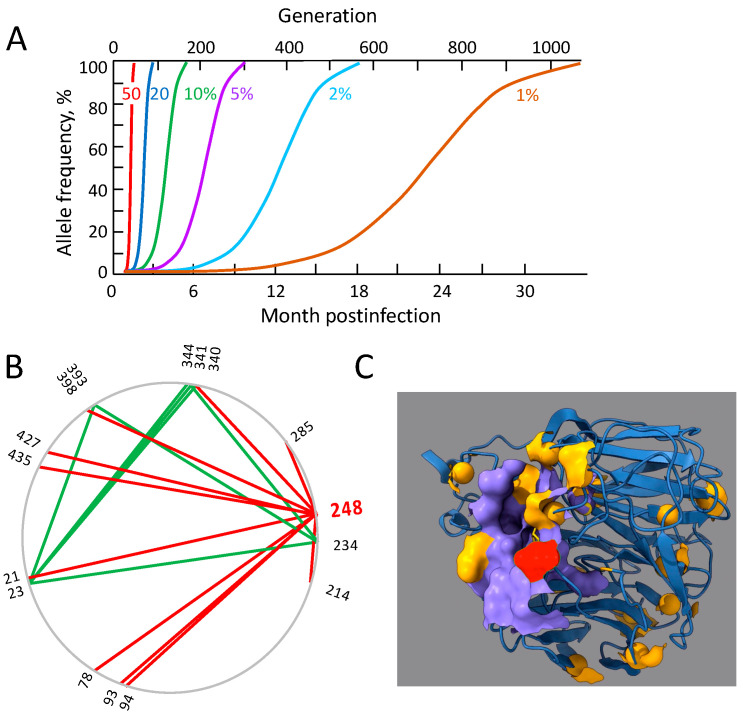
The fitness valley hypothesis. (**A**) Consecutive fixation of alleles, compensating for the fitness effect of the primary immune escape mutation in an HIV-infected patient. The percent selection coefficients are shown on the curves. (**B**) An example of an epistatic network predicted from genomic sequence data for the surface protein sequences of Influenza A H1N1, to explain the pandemics of 2009 [40]. The circular diagram shows the network of interaction between variable amino acid sites in the neuraminidase protein, as inferred by the 3-site haplotype method. The red lines show residue 248 with the primary mutation and the direct compensatory residues. The green lines show secondary compensatory mutations. (**C**) The structural location of the predicted epistasis network in a neuraminidase monomer of influenza virus. The three-dimensional structure of Influenza A H1N1 neuraminidase (PDB ID code 4QVZ). The colored spheres represent the predicted epistatic residues from (**B**). Red sphere: Predicted primary mutation, residue 248 in (**B**). Yellow spheres: Compensatory residues from (**B**). Note that all compensatory residues are located in alpha helixes on the surface of protein, representing a case of allosteric interaction. Based on [40].

**Figure 3 viruses-17-00197-f003:**
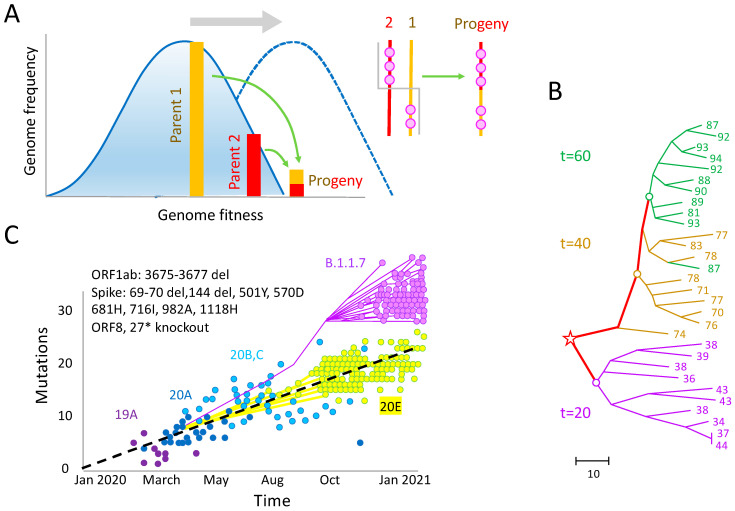
Multi-locus evolution with recombination. (**A**) The recombination of two viruses (parents, yellow and red bars) sometimes generates a highly fit recombinant virus (progeny), which extends the edge of genome distribution (blue solid curve) in fitness. It becomes the dominant strain later, when the distribution of genomes over fitness, the travelling fitness wave, moves over (gray arrow, blue dashed curve). Inset: the pink dots are beneficial alleles in two parental genomes that recombine into the progeny. (**B**) A phylogenetic tree for 10 genomes sampled at three time points from a population, shown in generations in 3 colors, the evolution of which is simulated by the Monte Carlo method (Fisher–Wright process). Th open circles show the roots of the three subtrees, which correspond to the previously best-fit genomes. The tree was obtained by neighbor-joining analysis in the MEGA11 package (version 11.0.10). The red star shows the root of the tree, chosen from the sampled sequences by MEGA11. The numbers at the leaves show the number of beneficial alleles in the genome. The genetic distance scale is below. Simulation parameters: population size N=1000, outcrossing rate r=1, locus number L=500, selection coefficient $s_{0}=0.1,$ initial allele frequency $f_{0}=0.02,\;M=3$ crossovers. (**C**) The dependence of the mutation number on time for various clades of SARS-CoV-2 during the formation of the VOC Alpha (B.1.1.7). Phylogeny is shown for two clades competing in UK in the Fall 2020, clade 20E (B.1.177) and variant B.1.1.7. (**B**,**C**) is based on [74,75], respectively.

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
