# Peer review of "Evolutionary Mechanisms of the Emergence of the Variants of Concern of SARS-CoV-2"

_viruses, 2025, doi:10.3390/v17020197_

Round 1

Reviewer 1 Report

Comments and Suggestions for Authors

The study presents two alternative hypotheses on the emergence of the variants of concern (VOC) of SARS-CoV-2. The prevailing view is that they arose from an epidemiological reservoir of immunocompromised patients not covered by genetic surveillance, where the virus evolves before returning into the main population.

Based on previous studies on HIV-1 and influenza virus, the two alternative hypotheses proposed by the author are respectively the "fitness valley" and the "multilocus evolution with recombination". In addition, the author suggests the adequate theoretical models to test the robustness of  both hypotheses. The paper, subdivided into 7 paragraphs,  has been carefully written. The cited bibliographic references are of high quality. 

I can only make a few suggestions:

Line 30, Introduction: can you better explain why "at least for the first two years"? The sense of "two years" reappears clearly in line 226, where it concerns the emergence of a VOC by Monte Carlo simulation. Put like this in Introduction, it may not be clear.

Line 40, Figure 1. This is just a curiosity. Why HIV-1 values are multiplied by 3?

Line 144, Figure 2. In population genetics, the range of the selection coefficient is from 0 to 1. It could be corrected as "percent selection coefficients".

Figure 3C is very interesting. Could it be explained more clearly in the legend or in the text? 

Line 270, References. The paper is present in database BioRXiv and the authors are Oversti, Gaul, Jensen, and Kuhnert.

The last point is a personal consideration that could perhaps be useful in the analysis of compensatory amino acid substitutions. In a study just submitted for publication, we evaluated the statistical significance of the covariation  between the substitutions in two viral proteins using the phi binomial correlation coefficient. Although I have never used it, I also know a multivariate statistical method, the spatial autocorrelation, that may be useful for the problem.  I just found in MEDLINE the paper "Spatial autocorrelation of amino acid replacement rates in the vasopressin receptor family", J Mol Evol, 2009, Lorraine Marsh. 

Author Response

Comment 1: The study presents two alternative hypotheses on the emergence of the variants of concern (VOC) of SARS-CoV-2. The prevailing view is that they arose from an epidemiological reservoir of immunocompromised patients not covered by genetic surveillance, where the virus evolves before returning into the main population.

Based on previous studies on HIV-1 and influenza virus, the two alternative hypotheses proposed by the author are respectively the "fitness valley" and the "multilocus evolution with recombination". In addition, the author suggests the adequate theoretical models to test the robustness of  both hypotheses. The paper, subdivided into 7 paragraphs,  has been carefully written. The cited bibliographic references are of high quality. 

Response 1: I thank the Reviewer for the positive view of my work.

Comment 2: I can only make a few suggestions: Line 30, Introduction: can you better explain why "at least for the first two years"? The sense of "two years" reappears clearly in line 226, where it concerns the emergence of a VOC by Monte Carlo simulation. Put like this in Introduction, it may not be clear.

Response 2: Thank you for pointing this out. This detail is obviously confusing and not of the primary importance for the central message of the manuscript. Hence I corrected the sentence in Line 20 to “Phylogenetic analysis demonstrates that a VOC did not descend directly from any of the current dominant virus variants, but represents a sister strain, with a common ancestor that existed many months ago”  and the sentence in line 226 to “If the estimated outcrossing rate is too low, and a Monte Carlo simulation based on it does not predict the emergence of VOCs with the frequency similar to that observed in real data, the hypothesis is falsified.”

Comment 3: Line 40, Figure 1. This is just a curiosity. Why HIV-1 values are multiplied by 3?

Response 3: Because HIV evolution is so much faster than that of other viruses, I had to rescale its values down by 1/3. Otherwise, the values for other viruses would be too low to read easily. The sentence now reads:  “To fit into the plot, HIV values are shown at a third of their actual value.”

Comment 4: Line 144, Figure 2. In population genetics, the range of the selection coefficient is from 0 to 1. It could be corrected as "percent selection coefficients"

Done in line 148.

Comment 5: Figure 3C is very interesting. Could it be explained more clearly in the legend or in the text? 

Response 5: I added in line 201: “Initially, the genetic distance from the original virus strain increased linearly with time. When variant Alpha appeared in the autumn of 2021, it deviated from this gradual increase by exhibiting a jump by 10 distinct mutations, including 3 short deletions, one knockout, and 6 aminoacid replacements (Fig 3C). Phylogeny hinted at a common ancestor of Alpha and the previous strain somewhere around March. Simulation (Fig. 3B) reproduces these features qualitatively including independent subtrees at different and even the same moments of time and jumps in the mutation number.”

Line 270, References. The paper is present in database BioRXiv and the authors are Oversti, Gaul, Jensen, and Kuhnert.

I already had this reference 2 in Intro and now added it to Line 62: “Several studies found an accelerated substitution rate within some chronic individuals compared to the virus in the main population 25-30, although some have challenged it 2.”

Comment 6: The last point is a personal consideration that could perhaps be useful in the analysis of compensatory amino acid substitutions. In a study just submitted for publication, we evaluated the statistical significance of the covariation  between the substitutions in two viral proteins using the phi binomial correlation coefficient. Although I have never used it, I also know a multivariate statistical method, the spatial autocorrelation, that may be useful for the problem.  I just found in MEDLINE the paper "Spatial autocorrelation of amino acid replacement rates in the vasopressin receptor family", J Mol Evol, 2009, Lorraine Marsh. 

Response 6: Thank you. I have added this last reference in line 159.

Reviewer 2 Report

Comments and Suggestions for Authors

Line 22: I am unsure of what (0 6 1 6) means in regards to the mutation rate of SARS-CoV-2. Are these the various mutation rates of the different proteins of the virus?

Figure 1 and Line 40: Are the shown bars on the figure already multiplied by 3, or are they to be multiplied by 3 to convey the proper mutation rates? Why do these values need to be multiplied by 3?

Line 64-65: I feel a citation demonstrating that evolution is faster in chronically infected individuals is required.

Line 172: Please expand on the indirect evidence supportive of the fitness valley hypothesis in SARS-CoV-2

Line 184-185: How is the view supported when the models have not yet been applied to SARS-CoV-2? Have these studies been applied to other coronaviridae?

I would like to see a perspectives section on what specific knowledge gaps remain in the emergence of SARS-CoV-2 VOCs, and how these gaps can be addressed. Additionally, based on the current data, are any of the hypotheses presented in this manuscript more likely than the others? What would these evolutionary mechanisms mean for potential of new VOCs to arise in the next few years? As infection prevention and control methods are applied to SARS-CoV-2, how would this evolutionary pressure affect any of the hypotheses presented in this manuscript?

Author Response

Comment 1: Line 22: I am unsure of what (0 6 1 6) means in regards to the mutation rate of SARS-CoV-2. Are these the various mutation rates of the different proteins of the virus?

Response 1: Thank you for this question. Clarified: “Depending on subtype”

Comment 2: Figure 1 and Line 40: Are the shown bars on the figure already multiplied by 3, or are they to be multiplied by 3 to convey the proper mutation rates? Why do these values need to be multiplied by 3?

Response 2: Clarified in Line 41: “To fit into the plot, HIV values are shown at a third of their actual value.”

Comment 3: Line 64-65: I feel a citation demonstrating that evolution is faster in chronically infected individuals is required.

Response 3: It is already given in Line 63: “Several studies found an accelerated substitution rate within some chronic individuals compared to the virus in the main population 25-30…”

Comment 4: Line 172: Please expand on the indirect evidence supportive of the fitness valley hypothesis in SARS-CoV-2

Response 4: Clarified: “At the present moment, the observation of strong compensation in SARS-CoV-2 represents indirect evidence 43, because it proves that compensatory mutations required for fitness valley effect, exist for this virus.”

Comment 5: Line 184-185: How is the view supported when the models have not yet been applied to SARS-CoV-2? Have these studies been applied to other coronaviridae?

Response 5: I replaced “view” to “possibility”, which is what I wanted to say.

Other than that, the models of population genetics are remarkably universal and can be adapted and modified between different viruses based on specific data obtained for each virus. Constructed from a small set of universal evolutionary factors, they can predict a very diverse set of behaviors.

Comment 6: I would like to see a perspectives section on what specific knowledge gaps remain in the emergence of SARS-CoV-2 VOCs, and how these gaps can be addressed.

Response: The Perspective is focused on the specific knowledge gap: the evolutionary mechanism of VOC. Other knowledge gaps, such as the effects of vaccination, deserve a separate consideration (see, for example, Rouzine & Rozhnova 2023 Comm. Med.).

Comment 7: Additionally, based on the current data, are any of the hypotheses presented in this manuscript more likely than the others?

Response 7: Once we test these hypotheses, we will know for sure. But, based on preliminary studies of my team, my bet is on fitness valley.

Comment 8: What would these evolutionary mechanisms mean for potential of new VOCs to arise in the next few years?

Response 8: We need to know more to answer this important question. I added a sentence, as follows (Line 284): “Once one of these hypotheses about the evolutionary mechanism is confirmed, as described above, and the relevant parameters are estimated, the prediction of the future VOC in the next few years will become possible, in the probabilistic sense, by Monte Carlo simulation.”

Comment 9: As infection prevention and control methods are applied to SARS-CoV-2, how would this evolutionary pressure affect any of the hypotheses presented in this manuscript?

Response 9: The same. Clarified in Line 270: “An interesting question is whether the evolutionary pressure from infection prevention and control (lockdowns, vaccines, etc) affects the hypotheses presented in this manuscript. Because these hypotheses are poorly studied in the context of SARS CoV-2, and we do not know which one is true, only the future simulation and data analysis will be able to answer this question.”.

Reviewer 3 Report

Comments and Suggestions for Authors

Dear Author,

Thank you for your submission. I found your paper to be a well-written and accessible overview of the evolutionary mechanisms underlying the emergence of SARS-CoV-2 variants of concern (VOCs). Your writing style is clear. However, I believe the paper could benefit from the following enhancements to increase its relevance and impact for readers seeking more detailed and up-to-date insights:

  1. Focus on Recent Developments
    While the paper offers a solid foundation, its generality limits its utility for researchers looking for updated details on VOC developments, particularly from 2022–2024. Expanding the discussion to include recent genomic data and integrating findings from studies published in 2023–2024 would significantly enhance its value.
  2. Greater Detail on SARS-CoV-2 Variants
    The use of popular names (e.g., Alpha, Beta, Omicron) is clear but lacks precision. Referring to these variants by their Pango lineage nomenclature (e.g., BA.1, BA.2, BA.5) is critical, especially for Omicron, given its multiple lineages. This refinement would provide readers with a more accurate and nuanced understanding.
  3. Suggestions for Specific Improvements
    • Line 29: Clarify the term “previous virus” and use Pango nomenclature where possible.
    • Figure 1: Adjust the size of the figure for readability. The color labeled as “brown” appears orange; please revise.
    • Section 2: The discussion of epidemiological reservoirs is oversimplified. Animal reservoirs should not be directly compared to human ones, as cross-species jumps often require additional adaptation mutations.
    • Line 79–81: Rephrase for clarity and correct grammatical issues.
    • Line 93: “All hypotheses are easily testable” seems overly broad. Clarify whether these hypotheses would indeed address the origins of VOCs.
    • Box 1: The definition of “genetic drift” needs clarification. Is the explanation referring to the host or the virus? The current description appears inaccurate for the virus.
    • Figure 2c: Specify whether this represents the NA monomer or tetramer.
    • Line 166: Correct the spelling of “neuraminidase.”
    • Figure 3A: Clarify the meaning of the blue horizontal lines on the Parent 2 graph.
    • Figure 3B: Explain the significance of the red star on the phylogenetic tree.
    • Figure 3C: Standardize the nomenclature for SARS-CoV-2 lineages to use either Pango or Nextstrain consistently.
    • Line 200: Your description of methods to rule out recombination appears outdated. Consider incorporating recent data, as most VOCs, VOIs, and VUMs in 2023–2024 are recombinants with confirmed genetic evidence.
    • Line 228: The assertion that data are insufficient to explain VOC emergence via recombination is incorrect. Recent studies demonstrate that recombination is a primary driver for most VOCs and VOIs.

By addressing these points, your paper would offer a more precise, detailed, and current contribution to the field. I look forward to seeing an updated version that integrates these suggestions.

Author Response

Dear Author,

Thank you for your submission. I found your paper to be a well-written and accessible overview of the evolutionary mechanisms underlying the emergence of SARS-CoV-2 variants of concern (VOCs). Your writing style is clear.

Thank you for a positive opinion about my paper.

However, I believe the paper could benefit from the following enhancements to increase its relevance and impact for readers seeking more detailed and up-to-date insights:

Comment 1: Focus on Recent Developments. While the paper offers a solid foundation, its generality limits its utility for researchers looking for updated details on VOC developments, particularly from 2022–2024. Expanding the discussion to include recent genomic data and integrating findings from studies published in 2023–2024 would significantly enhance its value.

Response 1: Thank you for this comment. I added a sentence with recent citations in Line 279: “Genetic evidence demonstrates the importance of recombination in the formation of recent genetic variants of SARS-CoV-2 [80-84].”

Comment 2: Greater Detail on SARS-CoV-2 Variants. The use of popular names (e.g., Alpha, Beta, Omicron) is clear but lacks precision. Referring to these variants by their Pango lineage nomenclature (e.g., BA.1, BA.2, BA.5) is critical, especially for Omicron, given its multiple lineages. This refinement would provide readers with a more accurate and nuanced understanding.

Response 2: I have added Pango nomenclature in parentheses (Line 27).

Suggestions for Specific Improvements

Comment 3: Line 29: Clarify the term “previous virus” and use Pango nomenclature where possible.

Response 3: Clarified: “Phylogenetic analysis demonstrates that a VOC does not descend directly from any of the current dominant virus variants, but represents a sister strain, with a common ancestor that existed many months ago [10-13].”

Comment 4: Figure 1: Adjust the size of the figure for readability. The color labeled as “brown” appears orange; please revise.

Response 4: I decreased the figure size and replaced “brown” with “light-brown”, as a compromise. The color of orange is usually brighter.

Comment 5: Section 2: The discussion of epidemiological reservoirs is oversimplified. Animal reservoirs should not be directly compared to human ones, as cross-species jumps often require additional adaptation mutations.

Response 5: Sentence added in Line 54-56: “A difference between the animal reservoir hypothesis and the hypothesis of silent spread in a human population is a presumed evolutionary rate. Another difference is that cross-species jumps often require additional adaptation mutations.”

Comment 6: Line 79–81: Rephrase for clarity and correct grammatical issues.

Rephrased in Lines 84-88: “The difference between VOCs and the virus in chronic patients could be explained by an evolutionary conflict between two biological levels [32, 33]: the level of a chronically-infected host and the level of a population. The evolutionary conflict, however, does not require chronic infection and can also result from the parallel evolution at the two biological levels.”

Comment 7: Line 93: “All hypotheses are easily testable” seems overly broad. Clarify whether these hypotheses would indeed address the origins of VOCs.

Response 7: Clarified in Line 105: All hypotheses of this class are testable, and their confirmation or falsification would help with understanding the origins of VOCs.

Comment 8: Box 1: The definition of “genetic drift” needs clarification. Is the explanation referring to the host or the virus? The current description appears inaccurate for the virus.

Response 8: Clarified: “Genetic drift – random fluctuation in the progeny number of a virus genome measured in the number of infected cells for within-host evolution and the number of infected hosts for population-level evolution, respectively.”

Comment 9: Figure 2c: Specify whether this represents the NA monomer or tetramer.

Response 9: Clarified: “monomer”.

Comment 10: Line 166: Correct the spelling of “neuraminidase”

Response 10: Corrected in Line 189.

Comment 11: Figure 3A: Clarify the meaning of the blue horizontal lines on the Parent 2 graph.

Response 11: I do not see any horizontal blue lines in the figure. Instead, I clarified the meaning of blue curves, both solid and dashed.

Comment 12: Figure 3B: Explain the significance of the red star on the phylogenetic tree.

Response 12: Explained in the caption: “The tree is obtained by neighbor-joining analysis in MEGA11 package. The red star shows the root of the tree chosen from the sampled sequences by MEGA11.”

Comment 13: Figure 3C: Standardize the nomenclature for SARS-CoV-2 lineages to use either Pango or Nextstrain consistently.

Response 13: Unfortunately, I have not found the Pango notation for variants 20A, 20B, C, only for 20E, which I added in the text and the caption.

Comment 14: Line 200: Your description of methods to rule out recombination appears outdated. Consider incorporating recent data, as most VOCs, VOIs, and VUMs in 2023–2024 are recombinants with confirmed genetic evidence.

Response 14: I researched the literature and added the sentence with a few recent citations in Line 279: “Genetic evidence demonstrates the importance of recombination in the formation of recent genetic variants of SARS-CoV-2 [80-84]. “

Comment 15: Line 228: The assertion that data are insufficient to explain VOC emergence via recombination is incorrect. Recent studies demonstrate that recombination is a primary driver for most VOCs and VOIs.

Response 15: I beg to differ. Three decades of work on virus evolution taught me that genomic data do not speak for themselves, as far as evolutionary mechanisms are concerned. Clarified in Line 281: “However, the fact that new strains are recombinants does not prove that recombination is the primary driver of their emergence, only that it participates in their formation. The most important evolutionary factor can be hiding behind the scene. Only performing computer simulation for alternative dynamic models and comparing their predictions to genomic data can determine, which evolutionary mechanism is the primary driver [56, 78, 85-87].”

By addressing these points, your paper would offer a more precise, detailed, and current contribution to the field. I look forward to seeing an updated version that integrates these suggestions.

Thank you for your time and effort.

Round 2

Reviewer 2 Report

Comments and Suggestions for Authors

I am satisfied with the author's revisions and responses to my concerns

Reviewer 3 Report

Comments and Suggestions for Authors

Dear author, 

Thank you for revision. All the points were addressed.